# Phylogenetic and Phylogeographic Analysis of the Highly Pathogenic H5N6 Avian Influenza Virus in China

**DOI:** 10.3390/v14081752

**Published:** 2022-08-11

**Authors:** Hanlin Liu, Changrong Wu, Zifeng Pang, Rui Zhao, Ming Liao, Hailiang Sun

**Affiliations:** 1College of Veterinary Medicine, South China Agricultural University, Guangzhou 510642, China; 2Key Laboratory of Zoonosis Control and Prevention of Guangdong Province, South China Agricultural University, Guangzhou 510642, China; 3National and Regional Joint Engineering Laboratory for Medicament of Zoonosis Prevention and Control, South China Agricultural University, Guangzhou 510642, China; 4Institute of Animal Health, Guangdong Academy of Agricultural Sciences, Guangzhou 510640, China; 5Key Laboratory for Prevention and Control of Avian Influenza and Other Major Poultry Diseases, Ministry of Agriculture and Rural Affairs, Guangzhou 510640, China

**Keywords:** H5N6 avian influenza virus, 2.3.4.4b reassortant, evolution, phylogeographic analysis

## Abstract

The clade 2.3.4.4b H5N8 avian influenza viruses (AIVs) have caused the loss of more than 33 million domestic poultry worldwide since January 2020. Novel H5N6 reassortants with hemagglutinin (HA) from clade 2.3.4.4b H5N8 AIVs are responsible for multiple human infections in China. Therefore, we conducted an epidemiological survey on waterfowl farms in Sichuan and Guangxi provinces and performed a comprehensive spatiotemporal analysis of H5N6 AIVs in China. At the nucleotide level, the H5N6 AIVs isolated in the present study exhibited high homology with the H5N6 AIVs that caused human infections. Demographic history indicates that clade 2.3.4.4b seemingly replaced clade 2.3.4.4h to become China’s predominant H5N6 AIV clade. Based on genomic diversity, we classified clade 2.3.4.4b H5N6 AIV into ten genotypes (2.3.4.4bG1–G10), of which the 2.3.4.4bG5 and G10 AIVs can cause human infections. Phylogeographic results suggest that Hong Kong and Jiangxi acted as important epicentres for clades 2.3.4.4b and 2.3.4.4h, respectively. Taken together, our study provides critical insight into the evolution and spread of H5N6 AIVs in China, which indicates that the novel 2.3.4.4b reassortants pose challenges for public health and poultry.

## 1. Introduction

The genetics and antigens of H5 AIVs, which circulate in wild birds and poultry globally, are diverse [1]. To date, H5 AIVs have evolved into different phylogenetic clades (clade 0–9) [2], and clade 2.3.4.4 has further evolved into eight subclades (2.3.4.4a–2.3.4.4h) according to the World Health Organization’s naming system [3]. H5 AIVs with the HA of clade 2.3.4.4b have been detected in wild birds and poultry worldwide [4,5,6] and have caused the loss of more than 33 million domestic poultry across the globe (https://empres-i.apps.fao.org/; accessed on 28 April 2022). Novel H5N6 reassortants were reported as one of the dominant AIV subtypes in China, especially in ducks, during 2014–2016 [7]. The clade 2.3.4.4h viruses became the dominant H5N6 lineage in China during 2018–2020 through continually evolving [8,9]. H5N8 AIVs with the HA of clade 2.3.4.4b are responsible for a new wave of outbreaks in poultry and wild birds in January 2020 [10,11]. Currently, the novel H5N6 reassortants with the HA from clade 2.3.4.4b H5N8 AIVs emerged in waterfowl and have caused human infections in China [12,13]. In this study, we conducted an epidemiological survey on local waterfowl farms in Sichuan and Guangxi provinces and performed a comprehensive spatiotemporal analysis of H5N6 AIVs in China from 2017 to May 2022. Our findings provide novel insight into the latest spatiotemporal characteristics of H5N6 AIVs and highlight their potential threat to public health.

## 2. Materials and Methods

### 2.1. Virus and Sequence Preparation

Two H5N6 AIVs designated as A/duck/Sichuan/SS1/2021 and A/goose/Guangxi/GG1/2022 were isolated from lung samples of waterfowl farms in Sichuan and Guangxi. The whole genome was amplified using universal primers [14]. All viral experiments were conducted in Biosafety Level 3 (BSL-3) facilities. All available H5N6 sequences isolated in China from 2017 to May 2022 were retrieved from the Global Initiative on Sharing All Influenza Data databases (GISAID; https://www.gisaid.org; accessed on 15 May 2022). Sequences with (a) evidence of lab errors and (b) 100% similarity were discarded. Detailed information on all H5N6 AIVs analysed in this study can be found in Appendix A.

### 2.2. Maximum Likelihood Phylogenetic of the H5N6 AIVs

Sequence alignments were constructed for eight segments separately using the MAFFT (version 7.149) program [15]. Using ModelFinder, the best-fit model was GTR+F+G4 [16]. Phylogenetic trees were inferred using the maximum likelihood method in the IQ-TREE 1.68 software with 1000 bootstraps [17,18]. We used ITOL (version 5) to complete the annotation of the evolutionary tree and to adjust it aesthetically [19]. The genotypes of the H5N6 AIVs were classified according to the combinations of lineages in segment trees (Appendix A) [7,12,18].

### 2.3. Bayesian Maximum Clade Credibility Phylogenetic and Demographic Analysis of the H5N6 AIVs

We used the TempEst (version 1.5.3) software to assess the temporal signals by performing root-to-tip regression analysis on viruses from clades 2.3.4.4b and 2.3.4.4h [20], which showed strong temporal signals suitable for demographic history analysis (Figure 1a,b). We computed marginal likelihoods using path sampling and stepping-stone sampling to compare the constant-size, exponential-growth and Bayesian skyline coalescent tree priors [21], and to compare the strict molecular clock and uncorrelated lognormal relaxed clock [22]. The best-fit model was chosen to construct Bayesian maximum cluster credibility (MCC) trees, utilising 300,000,000 total steps for each set, sampling every 1000 steps (Appendix A). The convergence (effective sample sizes > 200) of relevant parameters was assessed using Tracer (version1.7) [23]. TreeAnnotator (version 1.10.4) software was used to summarize the information from a sample of trees produced by BEAST on a single “target” tree to obtain the MCC tree. The phylogenetic tree was visualised in FigTree version 1.4.3 (http://tree.bio.ed.ac.uk/software/figtree/; accessed on 15 June 2022).

### 2.4. Phylogeographic Interference

We used the Bayesian stochastic search variable selection (BSSVS) model with asymmetric substitution to infer the H5N6 AIV spread dynamics from 2017 to May 2022 in China [24]. In order to reduce the sampling biases, we used CD-HIT v4.6 to remove viruses without known regional information and identical sequences [25], and further subsampled the dataset in a stratified manner to create the balanced number of sequences per region, which resulted in the final dataset including 28 viruses of clade 2.3.4.4b and 111 viruses of clade 2.3.4.4h (Appendix A). We considered transitions credible when the Bayes factor (BF) was >3 and used spreaD3 v0.9.6 to compute the BF tests to assess the support for significant individual transitions between distinct geographic regions [24,26]. We provide detailed information on BF values, migration rates and distances of clades 2.3.4.4b and 2.3.4.4h H5N6 AIVs in Appendix A. 

## 3. Results and Discussion

Molecular analysis revealed that all H5N6 AIVs in this study possess the same polybasic amino acid motif of -RRKR/GLF- in their HA cleavage site, which is a highly pathogenic avian influenza virus (HPAIV) characteristic (Appendix A). It is noteworthy that all H5N6 AIVs contained more than one of the seven HA amino acid mutations (94N, 133A, 154D, 155N, 156A, 188I and 189R), which could increase virus binding to α2, 6-linked sialic acid receptors [27,28,29,30,31]. Furthermore, most H5N6 AIVs contain the M1 15I, NS1 103F and NS1 106M mutations associated with virulence, transmission, replication efficiency and adaptation in mammals [32,33,34]. At the nucleotide level, two viruses isolated in our study exhibited high homology with those that cause human infections in the same provinces. The nucleotide homology for the eight genes was 98.88–99.60% between A/duck/Sichuan/SS1/2021 and A/Sichuan/06681/2021 and 95.18–99.61% between A/goose/Guangxi/GG1/2022 and A/GX/guilin/11151/2021 (Table 1).

We found that clades 2.3.4.4c, 2.3.4.4e and 2.3.4.4g contained relatively few viruses (Appendix A). Therefore, we used a virus branch to show the population of clades 2.3.4.4c and 2.3.4.4g and a histogram to represent the population of clade 2.3.4.4h (Figure 1c). Meanwhile, the estimated effective population size showed a sharp expansion of clade 2.3.4.4b H5N6 AIVs in 2020–2021, which is consistent with estimations when a new wave of clade 2.3.4.4b H5N8 entered China [35], and a continuous decline of clade 2.3.4.4h H5N6 AIVs in 2019–2022 (Figure 1c). Therefore, clade 2.3.4.4b seemingly replaced clade 2.3.4.4h to become the predominant lineage in China. Remarkably, the cross-host barrier transmission events of clade 2.3.4.4b H5N6 AIV expanded its host range to at least six species, including humans (Figure 1c).

**Figure 1 viruses-14-01752-f001:**
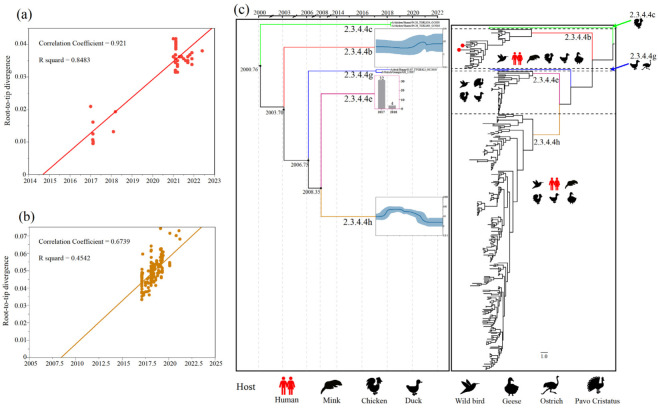
Population evolutionary relationships of the H5N6 AIVs in China during 2017–May 2022. (**a**) The temporal signal of clade 2.3.4.4b H5N6 AIVs root-to-tip regression analysis. (**b**) The temporal signal of clade 2.3.4.4h H5N6 AIVs root-to-tip regression analysis. (**c**) Classification and evolutionary history of H5N6 AIVs. (**Left**) Summary of coalescence analyses of the overall H5N6 sequences in China during 2017–May 2022, performed in BEAST (1.10.4). The blue curve inside the box is a Bayesian skyline plot (BSP), and the light blue curves represent upper and lower 95% highest probability density (HPD) values. The y axis of each BSP is relative genetic diversity, represented on a log10 scale. For some of the lineages (Clade 2.3.4.4c, 2.3.4.4e, and 2.3.4.4g), BSP was not performed because of the limited sample size. In the 2.3.4.4c and 2.3.4.4g lineages, we showed the branch of the viruses. In the 2.3.4.4e lineage, we used histograms instead, and the y-axis of each histogram shows the number of samples. The timeline of each lineage is aligned with the timeline of MCC tree. (**Right**) Overall phylogeny constructed using BEAST. The host range of each lineage was represented by animal icons.

Additional phylogenetic analyses of the internal genes of H5N6 AIVs revealed that the PB2, PB1, PA, NP, M and NS genes classified into five (Appendix A), five (Appendix A), six (Appendix A), four (Appendix A), five (Appendix A) and five (Appendix A) separate lineages, respectively. The internal genes of clade 2.3.4.4h H5N6 AIVs were mainly derived from the H6, H5N6 and H5N1 viruses, while the internal genes of clade 2.3.4.4b AIVs were primarily derived from the H5N8, H3N2, H5N1 and H7N9 viruses (Figure 2). Moreover, except for A/duck/China/0938/2017, almost all 2.3.4.4b H5N6 AIVs bore internal genes from the H5N8 virus (Figure 2). Therefore, the increasing genetic reassortment between H5N6 and H5N8 AIVs may pose an even greater threat to the poultry industry and public health. 

To further reveal H5N6’s evolutionary characteristics, we conducted molecular clock phylogenic analysis and genotype characterisation (Figure 2). Based on genomic diversity, clade 2.3.4.4b H5N6 AIVs were classified into ten different genotypes (2.3.4.4bG1–G10) (Figure 2), of which the 2.3.4.4bG5 and G10 AIVs caused human infections [12,13]. More specifically, 2.3.4.4bG5 viruses bore the NA gene from the Eurasian lineage, the PB2, PB1, PA and the NS genes from the H3N2 virus, the NP gene from the H5N1 virus and the M gene from the H5N8 virus. The 2.3.4.4bG10 viruses contain the NA gene from the Eurasian lineage, the PB2 and PB1 genes from the H3N2 virus, the NP and NS genes from the H5N1 virus and the M gene from the H5N8 virus (Figure 2; Appendix A). The genotypes of clade 2.3.4.4b H5N6 AIVs were most diverse in waterfowl, which are considered an essential reservoir for AIVs [36,37]. Moreover, 2.3.4.4b H5N6 AIVs exhibited distinct antigenicity to the Re-11 vaccine strain, which increases the risk to poultry [13]. 

Using a phylogeographic approach, we then estimated the migration pathways of clade 2.3.4.4b and 2.3.4.4h. There were 22 and 35 statistically significant migrations in clades 2.3.4.4b and 2.3.4.4h, respectively (Appendix A). The long-distance migrations of clades 2.3.4.4b and 2.3.4.4h occurred between Xinjiang and other regions (Figure 3a,d). The H5N6 AIVs from 2017 to May 2022 migrations of clades 2.3.4.4b and 2.3.4.4h also occurred between close regions (Figure 3b,e), and the transition rates between regions were inversely related to the distance (Figure 3c,f). These findings indicate that AIV migrations occur more frequently at shorter distances, consistent with a previous study [38]. The inferred spatial dynamics of H5N6 AIVs suggest that Hong Kong and Jiangxi acted as important epicentres of clades 2.3.4.4b and 2.3.4.4h H5N6 AIVs, respectively (Figure 3a,b). A previous study supports that the live poultry trade between different regions may facilitate the geographic migration of AIVs [7]. Currently, clade 2.3.4.4b H5N6 AIV is spreading in southern China (Figure 3a,b) and may further spread to other regions, which poses a challenge to public health. There were several limitations in this study. First, sampling bias may affect inference of the spread networks and association between the migration rate and distance. Second, the spatiotemporal HA datasets may not have accurately represented the times and locations of H5N6 AIVs originating in wild birds. Therefore, wild birds as long-distance vectors of HPAIV may result in an increased viral spread, a possibility that was not considered in our study [39,40].

In summary, this spatiotemporal analysis revealed that clade 2.3.4.4b has seemingly become the predominant H5N6 AIV in China. These viruses are genetically diverse among waterfowl and frequently spread in southern China. Moreover, the novel 2.3.4.4bG5 and 2.3.4.4hG10 reassortants can cause human infection, thus, posing a serious threat to public health. Hence, systematic surveillance of the H5N6 AIVs is critical for early warning and preparation for the next potential pandemic. 

## Figures and Tables

**Figure 2 viruses-14-01752-f002:**
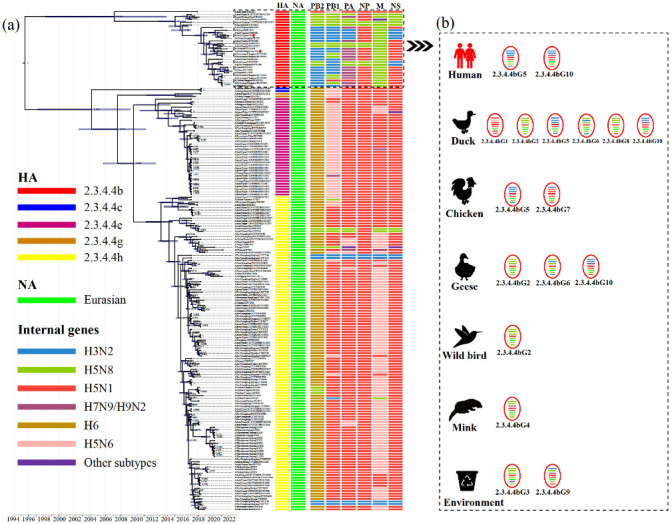
Molecular clock phylogenetic analysis of H5N6 AIVs in China during 2017–May 2022. (**a**) The phylogenetic tree of HA gene using BEAST (version 1.10.4) under the exponential coalescent tree prior model, “GTR+F+G4” substitution mode, and a “uncorrelated relaxed clock” model. The AIVs sequenced in this study were marked with red dots. The lineages of H5N6 AIVs genes were marked by coloured boxes. Purple node bars represented 95% credible intervals of lineage divergence times. (**b**) The genotype distribution of clade 2.3.4.4b H5N6 AIVs in different hosts. The genotypes of clade 2.3.4.4b H5N6 AIVs isolated in chickens, ducks, geese, minks, wild birds, environments, and humans. The eight segments represented by horizontal bars are, from top to bottom of the virion, PB2, PB1, PA, HA, NP, NA, M, and NS. Different colours represent different lineages. Detailed genotypes are available as the Appendix A. PB, polymerase basic protein; PA, polymerase acidic protein; HA, hemagglutinin; NP, nucleoprotein; NA, neuraminidase; M, matrix protein; NS, non-structural protein.

**Figure 3 viruses-14-01752-f003:**
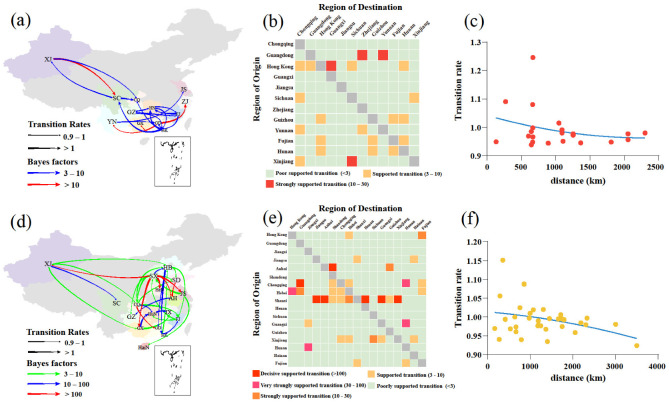
Phylogeographic analysis of clade 2.3.4.4b and 2.3.4.4h H5N6 AIVs was determined by Bayesian phylogeographic inference of HA in China during 2017–May 2020. Spatiotemporal spread of clade 2.3.4.4b H5N6 AIVs (**a**) and clade 2.3.4.4h H5N6 AIVs (**d**) curves show the inter-region virus transitions statistically supported with Bayes factor (BF) > 3; curve widths represent transition rate values; curve colours represent corresponding statistical BF for each transition rate. The heatmaps display the level of BF for each of the transition pathways considered for clade 2.3.4.4b H5N6 analyses (**b**) and 2.3.4.4h H5N6 analyses (**e**), respectively. Inter-region virus transition rates (BF >3) decrease with geodesic distance between different regions for clade 2.3.4.4b H5N6 AIVs (**c**) and clade 2.3.4.4h H5N6 AIVs (**f**), respectively. Abbreviations for the provinces are as follows: HeB: Hebei, CQ: Chongqing, HeN: Henan, SD: Shandong, JS: Jiangsu, HaiN: Hainan, ZJ: Zhejiang, GZ: Guizhou, HuN: Hunan, JX: Jiangxi, AH: Anhui, FJ: Fujian, GD: Guangdong, GX: Guangxi, SC: Sichuan, YN: Yunnan, HK: Hong Kong, XJ: Xinjiang, SX: Shanxi. The map in the square under Guangdong province indicates islands in the South China Sea.

**Table 1 viruses-14-01752-t001:** Nucleotide sequence identity between the virus isolated in this study with virus isolated in human available in GISAID ^a^.

Virus Isolated in This Study with Virus Isolated in Humans	Fragment ^b^	Identity (%)
A/duck/Sichuan/SS1/2021 with A/Sichuan/06681/2021	PB2	99.60
	PB1	99.46
	PA	99.39
	HA	99.58
	NP	99.39
	NA	95.49
	M	99.16
	NS	98.88
A/goose/Guangxi/GG1/2022 and A/GX/guilin/11151/2021	PB2	98.84
	PB1	99.33
	PA	99.23
	HA	99.41
	NP	99.39
	NA	95.18
	M	99.49
	NS	99.61

^a^ https://www.gisaid.org; accessed on 15 May 2022. ^b^ Polymerase basic subunit (PB); Polymerase acidic subunit (PA); Hemagglutinin (HA); Nucleoprotein (NP); Neuraminidase (NA); Matrix (M); Non-structural (NS).

## Data Availability

Consensus sequences generated in this study were submitted to the GISAID database, and their corresponding accession numbers are listed in Appendix A.

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
