# Peer review of "Phylogenetic and Phylogeographic Analysis of the Highly Pathogenic H5N6 Avian Influenza Virus in China"

_viruses, 2022, doi:10.3390/v14081752_

Round 1
Reviewer 1 Report
The present paper provides new perception into the latest characteristics of H5N6 AIVs and highlights their potential threat to public health. The study can be considered a valid short communication about the spatiotemporal distribution of H5N6 in China. In my opinion, only minor revisions are necessary to accepted the paper in this form.
Comments:
Lines 31 to 45: Maybe the introduction could de improved. Is there any other previous studies that describe the distribution of H5N6 in China in addition to those already mentioned?
Line 32: antigen should be replaced by antigens
Figure 1: Please replace "A, B and C" with "a,b and c". The Y-axis name is unreadable.
Figure 2: The phylogenetic tree is difficult to read. Is it possible to insert it as supplementary material?
Figure 3: The resolution of this figure is not particularly clear. The authors should improve the resolution to improve the quality.
Author Response
Responses to Reviewer 1 Comments
Dear reviewer,
Thank you for your comments made to our manuscript entitled “Spatiotemporal dynamics of the highly pathogenic H5N6 avian influenza virus in China” (Manuscript ID viruses-1855001). The authors have carefully reviewed the comments and revised the manuscript. The comments were very constructive and helpful in revising and improving our manuscript. The authors’ responses to the comments by the reviewers are listed as follows.
Best Regards,
Ming Liao, Ph.D.
Institute of Animal Health, Guangdong Academy of Agricultural Sciences, Guangzhou, China;
Email: mliao@scau.edu.cn
Lines 31 to 45: Maybe the introduction could be improved. Is there any other previous studies that describe the distribution of H5N6 in China in addition to those already mentioned?
Response 1: Thanks for your suggestion and question. The introduction was improved based on your suggestion by adding references related to distribution of H5N6 in China (Lines 36‒39).
Line 32: antigen should be replaced by antigens
Response 2: Thanks for your suggestion. “Antigen” has been replaced by “antigens” (Line 30).
Figure 1: Please replace "A, B and C" with "a, b and c". The Y-axis name is unreadable.
Response 3: Thanks for your suggestion. We have improved the resolution of Figure 1, and “A, B and C” have been changed into “a, b and c” (Lines 174 and 175).
Figure 2: The phylogenetic tree is difficult to read. Is it possible to insert it as supplementary material?
Response 4: Thanks for your suggestion. We have improved the resolution of Figure 2 to make the phylogenetic tree readable (Line 186).
Figure 3: The resolution of this figure is not particularly clear. The authors should improve the resolution to improve the quality.
Response 5: Thanks for your suggestion. We have increased the resolution to improve the quality of Figure 3 (Line 199).
Please see the attachment for the revised manuscript

Reviewer 2 Report
The manuscript submitted by Hanlin Liu et al. describes the evolution of China’s IAVs from 2017 to May 2022. The article is instructive to prevent the outbreak of IAVs virus, but because the sampling object is largescale farms, not wild birds, it takes time intervals for wild diseases to mutate into poultry. However, there are some English language errors throughout the manuscript and extensive professional language editing is needed.
Major comments excluding language errors:
Point 1: In line 2, Inappropriate title. The theme of the manuscript focuses on the molecular evolution analysis of avian influenza virus genes, not spatiotemporal dynamics.
Point 2: In line 4, No description for † symbol.
Point 3: In line 66, please introduce the detailed information of TempEst software, such as version, country, etc.
Point 4: In line 93-97, if these data are presented in tables, it is clearer than description.
Point 5:.In line 130, (Figure 2) should be deleted.
Point 6: In line 143, might should be revised to may.
Point 7: In Line 153, the title for Fig 1 is missing.
Point 8: The qualities of the pictures are not good. The marks in Figure 1 and Figure 3 are not clear, which need further clarification.
Author Response
Response to Reviewers 2 Comments
Dear reviewer,
Thank you for your comments made to our manuscript entitled “Spatiotemporal dynamics of the highly pathogenic H5N6 avian influenza virus in China” (Manuscript ID viruses-1855001). The authors have carefully reviewed the comments and revised the manuscript. The comments were very constructive and helpful in revising and improving our manuscript. The authors’ responses to the comments by the reviewers are listed as follows.
Best Regards,
Ming Liao, Ph.D.
Institute of Animal Health, Guangdong Academy of Agricultural Sciences, Guangzhou, China;
Email: mliao@scau.edu.cn
Point 1: In line 2, Inappropriate title. The theme of the manuscript focuses on the molecular evolution analysis of avian influenza virus genes, not spatiotemporal dynamics.
Response 1: Thanks for your suggestion. We have changed the title into “Phylogenetic and phylogeographic analysis of the highly pathogenic H5N6 avian influenza virus in China”(Lines 2 and 3).
Point 2: In line 4, No description for † symbol.
Response 2: Thanks for your suggestion. We have added the description of the † symbol (Line 12).
Point 3: In line 66, please introduce the detailed information of TempEst software, such as version, country, etc.
Response 3: Thanks for your suggestion. We have added detailed version information for TempEst (version 1.5.3) software (Line 68).
Point 4: In line 93-97, if these data are presented in tables, it is clearer than description.
Response 4: Thanks for your suggestion. These data have been presented in Table 1 (Line 166‒171).
Point 5:.In line 130, (Figure 2) should be deleted.
Response 5: Thanks for your suggestion. “Figure 2” has been deleted (Line 138).
Point 6: In line 143, might should be revised to may.
Response 6: Thanks for your suggestion. We changed “might” into “may” (Line 151).
Point 7: In Line 153, the title for Fig 1 is missing.
Response 7: Thanks for your suggestion. The title of Figure 1 has been added: Population evolutionary relationships of the H5N6 AIVs in China during 2017‒May 2022 (Line 173).
Point 8: The qualities of the pictures are not good. The marks in Figure 1 and Figure 3 are not clear, which need further clarification.
Response 8: Thanks for your suggestion. We have improved the resolution of Figures 1‒3 (Lines 172, 186 and 199 ).
Please see the attachment for the revised manuscript

Reviewer 3 Report
Title: Spatiotemporal dynamics of the highly pathogenic H5N6 avian influenza virus in China
This study conducted an epidemiological survey on waterfowl farms and performed a comprehensive spatiotemporal analysis of H5N6 avian influenza viruses (AIVs) in China. Moreover, this article reveals that clade 2.3.4.4b seemingly replaced clade 2.3.4.4h to become the predominant lineage in China and points out that the increasing genetic reassortment between H5N6 and H5N8 AIVs may pose an even greater threat to the poultry industry and public health.
This article provides novel insight into the latest spatiotemporal characteristics of H5N6 AIVs and highlights the potential threat of novel H5N6 reassortants to public health. It’s significant for the control of avian influenza viruses and preparedness for their potential threat. It’s recommended for publication, but some revisions need to be made.
Abstract
Line 17: Why did you choose Sichuan and Guangxi provinces to conduct the epidemiological survey? Just 2 provinces can’t represent China.
Keywords
Line 28: I suggest adding these keywords to make the paper easily searchable: H5N6 avian influenza virus; 2.3.4.4b reassortant; evolution; phylogeographic analysis.
Introduction
Line 40: I suggest revising “HA from H5N8 AIVs of 2.3.4.4b” to “HA from clade 2.3.4.4b H5N8 AIVs”.
Materials and methods
Line 49: More detailed information should be provided about the type of sample, such as swab samples or lung samples.
Line 75: Software and references for visualizing and beautifying maximum clade credibility (MCC) trees should be added to the materials and methods presented here.
Line 78: The number of sequences used in the phylogeographic analysis should be indicated here.
Line 79: Why migrations with Bayes factors (BFs) > 3 are considered credible should be supported by adding references here.
Results and Discussion
Line 86: Here the HPAIV abbreviation appears for the first time and the full name should be used. Highly pathogenic avian influenza virus (HPAIV).
Lines 94 and 95: Here, the abbreviation of influenza fragments first appeared in the manuscript, where the full name of the fragments should be added: polymerase basic protein (PB); polymerase acidic protein (PA); nucleoprotein (NP); neuraminidase (NA); matrix protein (M); non-structural protein (NS).
Line 131: References should be placed at the end of the sentence.
Line 136: Delete the “2.3.4.4h”.
Line 153: Figure 1 lacks a general heading. A title needs to be added to describe Figure 1 in general.
Lines 153 and 154: (A), (B) and (C) should be modified to (a), (b) and (c) so that the figure notes are written in accordance with the figure.
Line 187: Figure notes should be consistent with the picture, and (d) should be revised to (f).
Author Response
Response to Reviewer 3 Comments
Dear reviewer,
Thank you for your comments made to our manuscript entitled “Spatiotemporal dynamics of the highly pathogenic H5N6 avian influenza virus in China” (Manuscript ID viruses-1855001). The authors have carefully reviewed the comments and revised the manuscript. The comments were very constructive and helpful in revising and improving our manuscript. The authors’ responses to the comments by the reviewers are listed as follows.
Best Regards,
Ming Liao, Ph.D.
Institute of Animal Health, Guangdong Academy of Agricultural Sciences, Guangzhou, China;
Email: mliao@scau.edu.cn
Abstract
Line 17: Why did you choose Sichuan and Guangxi provinces to conduct the epidemiological survey?
Response 1: Thanks for your question. There were human cases of avian influenza in Sichuan and Guangxi provinces, and we found suspected symptoms of avian influenza in local waterfowl, so we conducted epidemiological investigations in these two provinces.
Keywords
Line 28: I suggest adding these keywords to make the paper easily searchable: H5N6 avian influenza virus; 2.3.4.4b reassortant; evolution; phylogeographic analysis.
Response 2: Thanks for your suggestion. The “Keywords” were changed into H5N6 avian influenza virus; 2.3.4.4b reassortant; evolution; phylogeographic analysis (Line 27).
Introduction
Line 40: I suggest revising “HA from H5N8 AIVs of 2.3.4.4b” to “HA from clade 2.3.4.4b H5N8 AIVs”.
Response 3: Thanks for your suggestion. “HA from H5N8 AIVs of 2.3.4.4b” was changed into “HA from clade 2.3.4.4b H5N8 AIVs” (Line 41).
Materials and methods
Line 49: More detailed information should be provided about the type of sample, such as swab samples or lung samples.
Response 4: Thanks for your suggestion. The viruses were isolated from lung samples of waterfowl farms in Sichuan and Guangxi (Line 50).
Line 75: Software and references for visualizing and beautifying maximum clade credibility (MCC) trees should be added to the materials and methods presented here.
Response 5: Thanks for your suggestion. Software and references for visualizing and beautifying maximum clade credibility (MCC) trees should be added (Lines 78‒80).
Line 78: The number of sequences used in the phylogeographic analysis should be indicated here.
Response 6: Thanks for your suggestion. The final dataset including 28 viruses of clade 2.3.4.4b and 111 viruses of clade 2.3.4.4h was used for our detailed phylogeographic analysis (Lines 87 and 88).
Line 79: Why migrations with Bayes factors (BFs) > 3 are considered credible should be supported by adding references here.
Response 7: Thanks for your suggestion. Reference has been added at the end of the sentence (Line 91).
Results and Discussion
Line 86: Here the HPAIV abbreviation appears for the first time and the full name should be used. Highly pathogenic avian influenza virus (HPAIV).
Response 8: Thanks for your suggestion. The full name of the highly pathogenic avian influenza virus (HPAIV) has been added (Lines 96 and 97).
Lines 94 and 95: Here, the abbreviation of influenza fragments first appeared in the manuscript, where the full name of the fragments should be added: polymerase basic protein (PB); polymerase acidic protein (PA); nucleoprotein (NP); neuraminidase (NA); matrix protein (M); non-structural protein (NS).
Response 9: Thanks for your suggestion. The full name of the fragments has been added (Lines 170 and 171).
Line 131: References should be placed at the end of the sentence.
Response 10: Thanks for your suggestion. The reference has been placed at the end of the sentence (Line 139).
Line 136: Delete the “2.3.4.4h”.
Response 11: Thanks for your suggestion. “2.3.4.4h” has been removed (Line 144).
Line 153: Figure 1 lacks a general heading. A title needs to be added to describe Figure 1 in general.
Response 12: Thanks for your suggestion. The title of Figure 1 has been added‒Population evolutionary relationships of the H5N6 AIVs in China during 2017‒May 2022 (Line 173).
Lines 153 and 154: (A), (B) and (C) should be modified to (a), (b) and (c) so that the figure notes are written in accordance with the figure.
Response 13: Thanks for your suggestion. (A), (B) and (C) have been mended to (a), (b) and (c) as suggested (Lines 174 and 175).
Line 187: Figure notes should be consistent with the picture, and (d) should be revised to (f).
Response 14: Thanks for your suggestion. (d) has been revised to (f) (Line 208).
Please see the attachment for the revised manuscript.
